# Metabolomic Profile of Indonesian Betel Quids

**DOI:** 10.3390/biom12101469

**Published:** 2022-10-13

**Authors:** Pangzhen Zhang, Elizabeth Fitriana Sari, Michael J. McCullough, Nicola Cirillo

**Affiliations:** 1School of Agriculture and Food, Faculty of Veterinary and Agricultural Sciences, The University of Melbourne, Parkville, VIC 3052, Australia; 2Discipline of Dentistry, Department of Rural Clinical Science, La Trobe Rural Health School, La Trobe University, Bendigo, VIC 3550, Australia; 3Faculty of Dentistry, Universitas Padjadjaran, Sumedang 45363, Indonesia; 4Melbourne Dental School, The University of Melbourne, Carlton, VIC 3053, Australia

**Keywords:** betel quid, areca nut, metabolomics, alkaloids, benzenoids, terpenes

## Abstract

Consumption of areca nut alone, or in the form of betel quid (BQ), has negative health effects and is carcinogenic to humans. Indonesia is one of the largest producers of areca nuts worldwide, yet little is known about the biomolecular composition of Indonesian areca nuts and BQs. We have recently shown that phenolic and alkaloid content of Indonesian BQs exhibits distinct geographical differences. Here, we profiled for the first time the metabolomics of BQ constituents from four regions of Indonesia using non-targeted gas chromatography–mass spectrometry (GC–MS) analysis. In addition to well-known alkaloids, the analysis of small-molecule profiles tentatively identified 92 phytochemicals in BQ. These included mainly benzenoids and terpenes, as well as acids, aldehydes, alcohols, and esters. Safrole, a potentially genotoxic benzenoid, was found abundantly in betel (*Piper betle*) inflorescence from West Papua and was not detected in areca nut samples from any Indonesian region except West Papua. Terpenes were mostly detected in betel leaves and inflorescence/stem. Areca nut, husk, betel leaf, the inflorescence stem, and BQ mixture expressed distinctive metabolite patterns, and a significant variation in the content and concentration of metabolites was found across different geographical regions. In summary, this was the first metabolomic study of BQs using GC–MS. The results demonstrate that the molecular constituents of BQs vary geographically and suggest that the differential disease-inducing capacity of BQs may reflect their distinct chemical composition.

## 1. Introduction

Betel quid (BQ) is a chewing package that typically includes the areca nut (AN), betel (*Piper betle*) leaf (BL), and slaked lime, with or without tobacco. Variations are also known to exist, including the addition of betel stem and inflorescence (SI, also referred to as BS or BI), and various additives [1,2]. BQ and AN use worldwide is comparable to that of tobacco, alcohol, and caffeine [3]. Globally, approximately one in ten people chew BQ, with the vast majority of its consumption concentrated in the Pacific Islands, South Asian, and Southeast Asian countries [3,4]. In these regions, AN and BQ chewing are the main risk factors for malignant and potentially malignant disorders of the oral cavity (OPMD), including oral submucous fibrosis (OSF), a chronic and progressively debilitating disorder associated with an increased risk of oral cancer [5]. The use of areca nuts in various forms is very popular in Indonesia, and this country is one of the largest producers of AN worldwide according to the International Agency for Research on Cancer (IARC) [1]. Within Indonesia, the prevalence of OPMD varies among different regions and individuals living in West Papua have been shown to have a much higher risk of developing pre-malignant lesions compared to those from Jakarta and West Java [6]. This may be due to the variation in the chemical composition of BQ consumed in different regions [7]. In our recent study investigating the chemical composition of Indonesian BQ mixtures, the highest arecoline concentration was observed in the nuts and BQ mixtures from West Papua; BQ mixtures from other regions of the country showed considerably lower concentrations of arecoline [7].

The mutagenic effects of constituents and extracts of the AN have been studied extensively in vitro and in vivo [8,9,10,11,12]. Four major AN alkaloids, namely arecoline, arecaidine, guvacine, and guvacoline, have been isolated and suggested to play a key role in oral carcinogenesis. However, AN is commonly consumed in a BQ mixture package that most commonly includes *Piper betle* leaf (betel leaf) or *Piper betle* stem inflorescence (betel SI) and slaked lime [13] (Appendix A). This is a crucial point in BQ research as the composition and biological activities of individual ingredients change significantly when these are combined into a BQ mixture [7]. For example, the addition of slaked lime, namely calcium hydroxide, facilitates the hydrolysis of arecoline to arecaidine, which leads to amplified fibroblast proliferation, increased collagen formation, and potentially an increased risk of oral cancer [14,15,16]. Betel SI containing safrole is likely to be carcinogenic [17,18,19]; however, it is not known whether this genotoxic benzenoid is equally abundant in the BQ mixture of different origins. Thus, while AN might not be the only factor responsible for the pathophysiological effects of BQs, it is important to investigate the biomolecular composition of BQ ingredients both individually and as a mixture.

The most widely reported method for identifying and quantifying alkaloids in AN has been high-pressure liquid chromatography (HPLC) and liquid chromatography–mass spectrometry (LC–MS) [20,21], with the most accurate method using isotope labelled standards and liquid chromatography-tandem mass spectrometry [22]. Gas chromatography–mass spectrometry (GC–MS) has also been used to identify alkaloid compounds in plant materials [23,24]. GC–MS is more commonly used to screen small molecules in plant and food samples [25]. Here, we decided to use GC–MS to profile nature molecules in BQ ingredients of different geographical origins. To the best of our knowledge, this was the first study attempting to employ a GC–MS platform to profile BQ ingredients, especially the *Piper betle* inflorescence and husk. The BQ ingredients assessed in this study were AN, betel leaf, betel SI, areca husk, and BQ mixture containing AN, betel leaf/SI, and slaked lime. These BQ samples were sourced from four different geographic regions of Indonesia, namely Banda Aceh (BA), North Sumatra (NS), West Kalimantan (WK), and West Papua (WP).

## 2. Materials and Methods

### 2.1. Chemicals and Reagents

Arecoline hydrobromide and guvacine hydrochloride were purchased from Sigma-Aldrich, (Castle Hill, NSW, Australia), arecaidine hydrobromide and guvacoline hydrobromide were supplied by Sapphire Bioscience, Pty Ltd. (Redfern, NSW, Australia). Its purity and identity were checked by GC–MS and 1H-NMR. Methanol (HPLC grade), chloroform, sulfuric acid, and ammonia (analytical grade), 4-octanol, as the internal standard (IS), and pyridine, were purchased from Sigma–Aldrich (Castle Hill, NSW, Australia). The hydrocarbon mixture (C7–C30, Restek, cat no. 31614) was supplied by Teknokroma (Barcelona, Spain).

### 2.2. Experimental Design and Sample Preparation

BQ samples were collected from four regions of Indonesia: Banda Aceh (BA), North Sumatra (NS), West Kalimantan (WK), and West Papua (WP). The four regions selected represented the western, middle, and eastern parts of Indonesia. For BA, NS, and WK regions, dried AN, (Appendix A), areca husk (Appendix A), betel leaf, (Appendix A), betel inflorescence/stem (Appendix A), and mixed BQ samples [7] were analysed. The mixed BQ contains dried AN, betel leaf, and slaked lime (Appendix A) at the ratio of 80.5:12.5:7 by weight [11]. For the WP region, dried AN, areca husk, betel SI, and mixed BQ samples were analysed. Unlike other regions, the BQ from the WP region contains betel inflorescence stem commonly named as a flower, instead of a betel leaf [5].

All samples were freeze-dried for 72 h using FD3 Freeze Drier (Dynavac Engineering, Osborne Park, Australia), and finely ground using an electronic grinder (Multigrinder II, model EM0405, Sunbeam, Auckland, New Zealand). A total of 5 g of sample was mixed with 50 mL methanol and shaken for 24 h at room temperature. Samples were then centrifuged at 6500 rpm for 10 min to obtain the supernatant. The supernatant was concentrated at 40 °C using a Hei-VAP value rotary evaporator (Heidolph Instruments GmbH & Co.KG, Schwabach, Germany) and topped up to 10 mL using methanol. The concentration step allowed the detection of compounds present in low concentrations. The final extract was flushed with nitrogen gas and sealed and stored at 4 °C in darkness until analysis. The extract solution was filtered through a 0.45 µ membrane. A total of 0.9 mL of sample was transferred to 2 mL of GC vial and mixed with 42 µL of 4-octanol as the internal standard (IS) (1 mg/L in methanol). All samples were prepared in triplicates. Alkaloids standards (arecoline hydrobromide, arecaidine hydrobromide, guvacoline hydrobromide, and guvacine hydrochloride) were accurately weighed into a volumetric measuring flask of 10 mL. A total of 40 μL of 30% ammonia was added and then diluted in methanol (with a range of concentrations of 100, 50, 25, 12.5, and 6.25 mg/mL).

### 2.3. Gas Chromatography-Mass Spectrometry (GC–MS) Analysis

GC–MS analysis was performed following the published method on alkaloids [26]. Briefly, an Agilent Technologies 6850 series II (GC; Agilent Technologies, Santa Clara, CA, USA) connected to an Agilent Technologies 6973 mass selective detector (MSD) and coupled with an Agilent PAL multipurpose autosampler was used for volatile analysis. The instruments were controlled using Agilent G1701EA MSC ChemStation software in conjunction with Agilent PAL autosampler Control software B.01.04 for ChemStation. The GC was fitted with a J&W DB-5 ms column (30 m × 0.25 mm, 0.25 µm film df, with helium as carrier gas (ultrahigh purity, BOC Australia, North Ryde, NSW, Australia). The GC inlet was fitted with a borosilicate glass split inlet liner (volume 935 µL, Agilent Technology) and held at 250 °C; 1 µL of the sample was injected into the inlet in the split mode at a split ratio of 30:1, and a constant column flow rate of 1 mL/min. The column temperature was initiated at 100 °C and increased to 180 °C at 15 °C/min, held at 180 °C for 1 min, then increased to 300 °C at 5 °C/min and held for 6 min and post-run at 100 °C for 2 min. The MS source, quadruple, and transfer line were held at 230 °C, 150 °C, and 280 °C, respectively. The MS was operated in scan/sim mode (35–350 m/z) with a positive EI of 70 eV. The sim target ions for alkaloids quantification include: 53, 55, 69, 73, 82, 96, 126, 127, 140, 141, 155 m/z. The standard solution of arecoline, arecaidine, guvacoline, and guvacine were analysed using the same method as samples. All standard solutions were spiked according to internal standards and analysed as samples to generate standard curves (R^2^ > 0.99). Alkaloids were identified by comparing the mass spectra and retention indices with the pure standard, while other metabolites in the sample were identified using the NIST library in ChemStation and Webbook database and the standard solutions obtained. All compounds were quantified based on the standard curve curves using target ions. Blank and internal standards were checked regularly to check the sensitivity of the GCMSD system.

### 2.4. Statistical Analysis

Statistical analysis was conducted using MS-Excel, Minitab 17© 2016, MATLAB© by Mathworks, XLSTAT (2019.3.2.61308), and R software (v3.5.2 The R foundation). One-way ANOVA (with the odds ratio test) was executed at *p* ˂ 0.05. The groups indicated with different letters showed statistically significant differences. The results are exhibited as the means + standard deviation (SD) from four observations of individual replicates.

## 3. Results

This study tentatively identified 92 compounds and confirmed 3 alkaloids in BQ ingredients. WP-SI showed the highest concentration of metabolites in almost all chemical groups examined, namely benzenoids, terpenes, and esters compared to other samples (Appendix A and Figure 1).

### 3.1. Benzenoids

There were 20 benzenoids identified in the current study using GC–MS. Safrole was one of the most abundant benzenoids detected. The highest concentration of safrole was measured in WP-SI, which was the main source of safrole for the BQ mixture from WP. Safrole was undetected in BA-AN, NS-AN, and WK-AN, and was detected at a very low concentration in WP-AN. Safrole was also detected in leaf samples from all regions. In the husk group, safrole was detected in all samples.

Eugenol has a structure similar to safrole and was also found in betel leaf and betel SI samples. WP-SI contains the highest eugenol levels compared to BA-leaf, NS-leaf, and WK-leaf. However, eugenol was not detected in the nut samples.

A considerably high concentration of 3-Allyl-6-methoxyphenol-acetate was found in WP-SI. Other benzenoids, such as 5-hydroxymethylfurfural, were identified in WP-husk and BA-husk. Moreover, 1,3,5-benzenetriol was detected in all betel leaf samples; however, it was not detected in WP-SI (Appendix A and Figure 1).

### 3.2. Terpenes

Our study showed that terpenes were mostly detected in betel SI and betel leaf groups. WP-SI contained the highest concentration of almost all terpene compounds detected especially sesquiterpenes such as γ-cadinene, spathulenol, α-copaene, α-gurjunene, and γ-amorphene; however, it had a lower concentration of phytol compared to leaves. Specifically, phytol was present in BA-leaf, NS-leaf, and WK-leaf. While D-limonene was detected in all samples, NS-AN contained the highest level of this molecule.

In total, 20 sesquiterpenes were detected in the samples, with the majority of them detected at higher concentrations in WP-SI samples. For instance, an appreciable concentration of spathulenol was identified in WP-SI samples as well as in betel leaf from WK, NS, and BA. A high concentration of γ-cadinene also was found distinctively in WP-SI compared to WK-leaf, NS-leaf, and BA-leaf (Appendix A and Figure 1).

### 3.3. Other Metabolites (Acid, Aldehyde, Alcohol, Ester)

The results of alcohol metabolites indicated the presence of 2,3-butanediol in NS-husk, NS-AN, NS mixture, and WK-husk. Glycerine was found mostly in husk samples with NS-husk containing the highest concentration compared to husk samples from other regions.

In the group of acids metabolites, L-lactic acid was found at the highest concentration in NS-husk, NS-AN, and BA-husk, but it was undetected in the remaining samples. Isovanillic acid was only detected in husk samples (all regions), whereas 9,12-octadecadienoic acid (Z,Z)- was detected in all samples but nuts.

The ester group of chemicals was mostly detected in husk and leaf samples. WP-husk contains the highest levels of hexadecanoic acid and linoleic acid ethyl ester. Methyl nicotinate was most abundant in WP-AN, compared to WK-AN, NS-AN, and BA-AN.

### 3.4. Alkaloids in Betel Quids

Using GC–MS device, 3 alkaloids standards, arecoline, arecaidine, and guvacoline, showed stable peaks, whereas guvacine was detected but demonstrated unstable peaks (Figure 2). Therefore, guvacine could not be quantified with this method. The mass spectrum for each alkaloid standard was very clear and could be used to identify these compounds present in BQ samples (Figure 3).

Arecoline was found in all samples, except in WP-SI and WK-leaf. In the AN group samples, arecoline was found at the highest concentration in WP-AN (12.0 mg/g DM), followed by BA-AN (7.2 mg/g DM), WK-AN (5.3 mg/g DM), and NS-AN (3.7 mg/g DM). Importantly, the abundance of arecoline in BQ mixtures followed the same geographical rank seen in the AN groups, albeit at lower concentrations. WP region also showed the highest arecoline levels in the husk group (5.0 mg/g DM) compared to other regions (with a range < 0.4 mg/g DM). The lowest concentrations of arecoline were detected in BA-leaf and NS-leaf both at 0.1 mg/g DM (Table 1).

Arecaidine was only detected in NS-husk and NS-mixture samples but not in any other samples. The concentration of arecaidine in the NS-husk was approximately 26 folds of that in NS-mixture (Table 1). However, the concentrations measured were above the range of standards prepared and, thus, not quantified.

Trace amounts of guvacoline were also detected in the BQ mixture from BA, NS, and WK, with the highest concentration detected in BA-mix at 1.1 mg/100 g DM. No guvacoline was detected in AN, husk, leaf, and BI samples. No guvacine was detected in any of the samples.

## 4. Discussion

In the present study, we used GC–MS to carry out the metabolomic profiling of BQ ingredients from four regions of Indonesia. This allowed us to identify, for the first time, a broad spectrum of biomolecules extracted from BQ ingredients. Of the 92 metabolites detected, several are known to be involved in the development of OSMF and oral cancer [10,17,27,28,29,30,31,32,33].

Benzenoids are the most abundant floral scent molecules and originate from the aromatic amino acid phenylalanine, which provides the characteristic benzene ring that is decorated by oxidation, acylation, or methylation to yield the individual scent components [34]. Previous studies have reported that betel SI contains a high concentration of safrole, eugenol, hydroxychavicol, isoeugenol, and eugenol methyl ester [5,35,36]. A study by Hwang et al. reported that safrole was the major phenolic compound detected by HPLC in Taiwan’s betel SI [37]. These data are consistent with our current results, where betel SI from WP contains a high concentration of safrole, eugenol, and some other phenolics.

Safrole is a potential human carcinogen [37,38] and, importantly, was one of the benzenoids that most abundantly present in samples from WP, a region with a high incidence of OSMF and oral cancer. A study using the nuclease P1 version of the ^32^P-postlabelling technique successfully determined safrole-DNA adducts in two out of 28 hepatic tissues from patients with hepatocellular carcinoma, and only these two patients had histories of more than 10 years of BQ chewing habits. These results suggest that BQ-containing safrole might be involved in the pathogenesis of hepatocellular carcinoma in humans [18]. A study by Chiu-Lan Chen et al. demonstrated that safrole forms stable safrole–DNA adducts in human oral tissues following BQ chewing, which may contribute to oral carcinogenesis [17].

Safrole was not only present in betel SI but also in areca husk samples. All areca husks tested contained safrole. Interestingly, our study found both arecoline and safrole in all areca husk samples. This finding might suggest that adding husk into BQ could further raise the risk of developing OSMF and oral cancer. However, it must be noted that there is some evidence that safrole and its derivatives may exert anti-proliferative and hence tumour-suppressing activity [39,40]. For example, safrole has been shown to induce apoptosis in human oral cancer cells [40].

It is important to point out that not all benzenoids are dangerous, and many have potentially beneficial effects on human health, such as antioxidant, antimicrobial [41], and inflammatory properties [42]. In vitro studies documented eugenol activity against breast and liver cancer [41,43]. The antimicrobial and antioxidant properties of eugenol are well known and are widely used as food additives and as pharmaceutical, cosmetic, and active packaging materials [44].

The terpenes family has over 30,000 members that are typically known as constituents of flavours, antifeedants, and pheromones [45,46]. Terpenes are made from the isoprene unit (C_5_), and, and classified as hemi- (C_5_), mono- (C_10_), sesqui- (C_15_), di- (C_20_), sester- (C_25_), tri- (C_30_), tetra- (C_40_), or polyterpenes ((C_5_)_n_; n > 8) based on the number of their unit (46). Terpenes have been reported to harbor anti-inflammatory, anti-carcinogenic, cardioprotective, and neuroprotective effects that benefit human health [46].

Our current study identified 24 terpenes with 20 of them being sesquiterpenes. Most terpenes were detected in betel SI and betel leaf groups, with WP-SI samples having the highest concentration of most compounds, such γ-cadinene, spathulenol, α-copaene, α-gurjunene, γ-amorphene (Appendix A). A similar study examining the composition of betel leaf var. *Bangla* essential oil using GC–MS identified several monoterpenes and sesquiterpenes reported in the current study such as linalool, chavicol, anethole, estragole, α-copaene, and caryophyllene. Those terpenes exhibited antibacterial activity against *Mycobacterium smegmatis, Staphylococcus aureus*, and *Pseudomonas aeruginosa* [36]. While previous studies were successful in identifying numerous metabolites from betel leaf, our study is the first to have ever profiled betel SI using GC–MS and identified compounds, especially sesquiterpenes.

Husk was shown to contain alcohol, acid, and ester group metabolites in our current study (Appendix A). The benefits of L-lactic acid as a probiotic have been widely reported. The wide ranges of lactic acid bacteria have been extensively studied for their ability to produce extracellular polysaccharides [47]. Employing extracellular polysaccharides has been suggested for future use in treating cancer and immune-related diseases [47]. L-Lactic acid was found at the highest levels in NS-husk but not detected in WP-husk.

It has been observed in several studies that the four main alkaloids found in AN are arecoline, arecaidine, guvacine, and guvacoline [48,49,50]. Our data confirmed the major alkaloid found in BQs to be arecoline, as previously reported [51]. We failed to detect a stable guvacine peak under our experimental conditions. This was most likely due to guvacine’s melting point (306–309 °C), which exceeds the inlet and oven temperature set (300 °C). Thus, this method is not suitable for profiling guvacine. The tailing effect was also observed in arecoline, arecaidine, and guvacoline peaks, which was possible due to the fact that we used the standards in the salt form in reaction with ammonia to produce the free alkaloids.

Arecoline was found in all samples, except in WP-SI and WK-leaf. In the AN samples group, arecoline was found at the highest concentration in WP-AN, followed by BA-AN, WK-AN, and NS-AN in descending order. The different levels of arecoline observed among areca nuts could be due to several factors such as the maturity of the nut, storage, and geographical location [22]. For example, arecoline level rises in the immature AN and drops significantly in the mature AN [52]. This is consistent with our results showing that immature WP-AN had a high concentration of arecoline, whereas this alkaloid was less abundant in mature BA-AN and NS-AN. In our experimental conditions, arecoline was detected, albeit at low concentrations (0.1 mg/g DM) in leaf samples from BA and NS, but not in WK-leaf. Previous studies did not detect alkaloids in betel leaves [21,36,53]. This variation may be due to the variation in the specific cultivar grown in different geographic regions and the climate condition, which need to be further validated in a future study.

In our study, arecoline was also present in the husk group from all regions, with husk from WP containing more than 10-fold higher levels of arecoline compared to BA, NS, and WK. Strikingly, people living in Taiwan, China, Guam, and Papua New Guinea [5] as well as the WP region [6] include the husk of the outer AN pericarp as part of the BQ mixture. Our finding, therefore, has salient clinical implications as it could explain how the husk added to BQ contributes to the high risk of OSMF and oral cancer in BQ chewers from WP. Consistent with this hypothesis is the finding that the WP mixture had the highest concentration of arecoline compared to mixtures from other regions. In our study, the level of arecoline was lower in the mixtures compared to AN alone, and this may occur due to the dilution effect of multiple ingredients, the potential interaction between BQ ingredients, and the overall chemical environment facilitating arecoline processing. This is extremely important for the pathogenesis of both OSMF and OSCC as an alkaline environment may promote fibrosis directly [54] and contribute to oral carcinogenesis via the production of mutagenic metabolites. For example, arecoline N-oxide, a metabolite of areca nut alkaloids, forms protein adducts that can be detected in oral keratinocytes transiently treated with areca nut extracts and induces an increase of fibrotic related genes including TGF-beta-1, S100A4, MMP-9, IL-6, and fibronectin and a decrease of E-cadherin compared with arecoline [55]. Further research is required to better understand the interaction of chemical constituents of BQ ingredients in different pH conditions.

Arecaidine was only detected in NS husk and mixture samples at low concentrations, while guvacoline was detected in mixture samples from BA, NS, and WK. Arecaidine and guvacoline were not detected in WP and WK mixtures. A study on aqueous AN extract by Franke et al. [27] on total alkaloid level reported that a green unripe AN contains a lower level of total alkaloids compared to the yellow ripe AN, with arecoline as the main alkaloid in unripe AN. This could explain why arecaidine and guvacoline were detected in riper samples from BA and NS regions and arecoline was the most abundant alkaloids in unripe WP-AN and WK-AN. The arecaidine detected in the NS mixture might indicate the hydrolysis of arecoline to arecaidine that takes place in high pH conditions [22], e.g., due to the addition of slaked lime. This could provide an additional pathogenic mechanism for the carcinogenic effect of slaked lime (calcium hydroxide) [56,57], given its ability to generate reactive oxygen species (recently reviewed in [58]). The level of arecaidine found by a study in pan masala and gutka ranged from 0.14 to 1.70 mg/g DM, while guvacoline was quantified in the range of 0.17 to 0.99 mg/g DM. These results are consistent with our data, and the slight differences observed are likely due to several factors such as the maturity of AN, the processing methods, and growing conditions [27].

## 5. Conclusions

Here, we successfully profiled 92 plant metabolites and 3 alkaloids in BQ compounds using GC–MS. Arecoline was the major alkaloid found in the AN and BQ mixture group, whereas WP-AN and WP mixtures were identified as having the highest levels of arecoline among other samples. Arecoline was also present in the husks and was particularly abundant in samples from WP. This suggests that the addition of husk in BQ mixtures is potentially hazardous. The benzenoid safrole was detected in betel SI at a very high concentration, whereas betel leaf and areca husk samples only contained small levels of this genotoxic biomolecule. Importantly, safrole was only identified, albeit at low concentrations in AN samples from WP, but was not detected in BA-AN, WK-AN, and NS-AN. Overall, our data strongly point to WP as the Indonesian region with potentially the most harmful BQ ingredients.

## Figures and Tables

**Figure 1 biomolecules-12-01469-f001:**
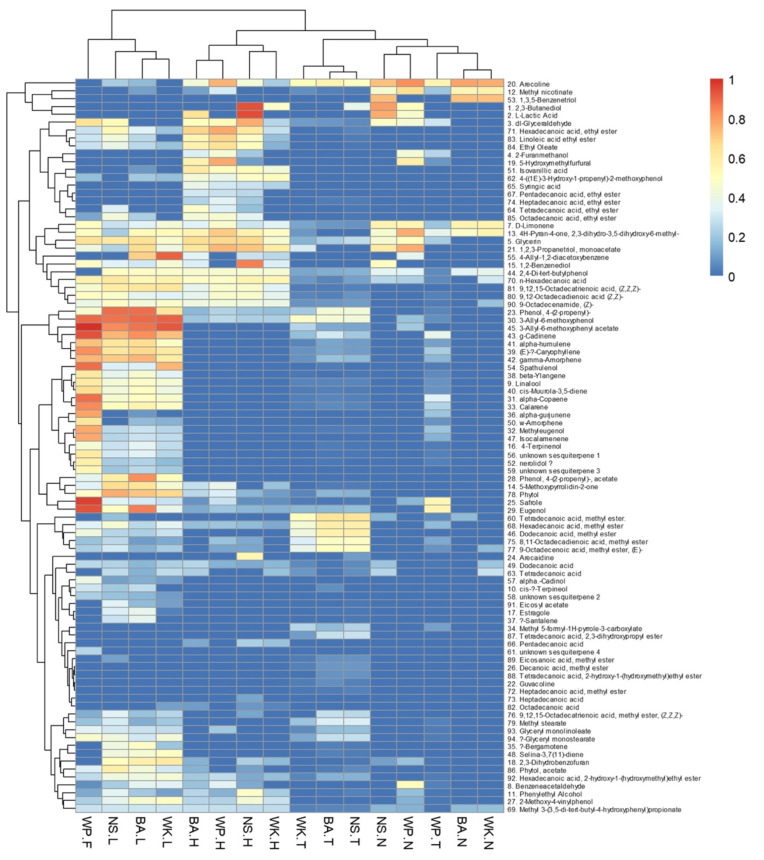
Heatmap of natural compounds found in betel quid ingredients.

**Figure 2 biomolecules-12-01469-f002:**
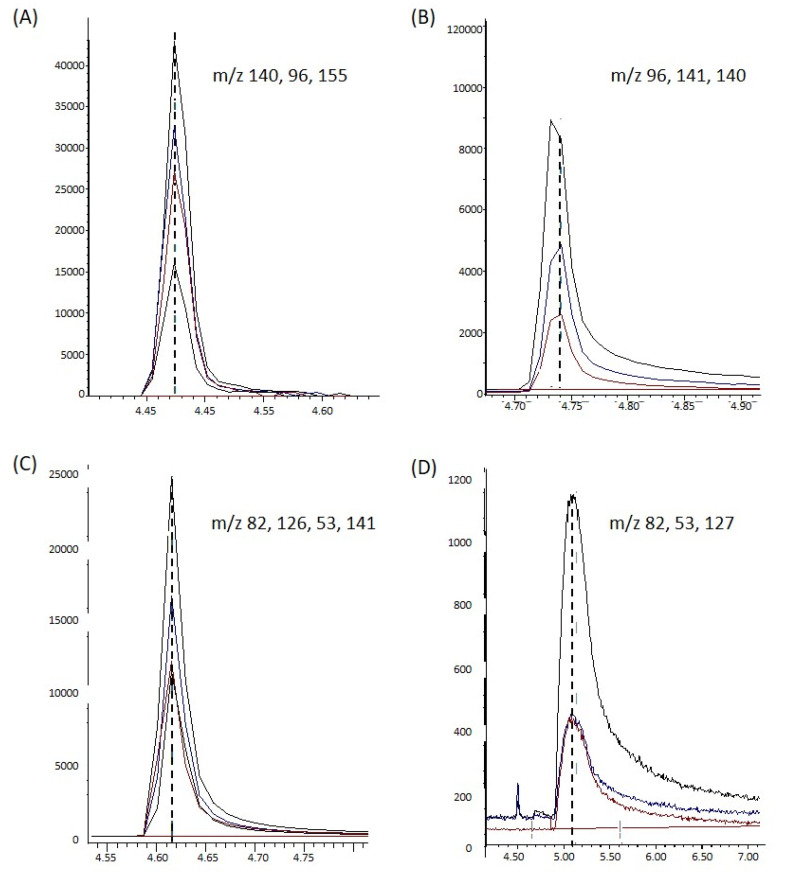
Peaks of alkaloids; (**A**) arecoline, (**B**) arecaidine, (**C**) guvacoline, and (**D**) guvacine.

**Figure 3 biomolecules-12-01469-f003:**
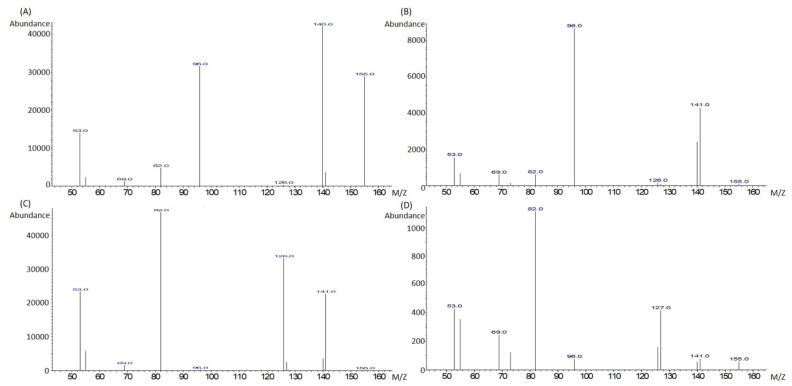
Mass spectrum of target ions of arecoline (**A**), arecaidine (**B**), guvacoline (**C**), and guvacine (**D**).

**Table 1 biomolecules-12-01469-t001:** Composition of alkaloids in Betel Quid determined by GC-MS.

Peak Number	RI	RI_db_	Chemicals	Target Ion	QC Ion	Retention Time (min)	Standard Curve Equation	R²	Range	BA				NS				WK				WP			
										Husk	Leaf	AN	Mix	Husk	Leaf	AN	Mix	Husk	Leaf	AN	Mix	Husk	SI	AN	Mix
1	1237.16	1236	Arecoline	140	155, 96, 81, 53, 124	4.493	y = 9.2899x^2^ + 149.17x + 7.4312	0.9971	6.25–1000 mg/L	0.3 ± 0.0	0.1 ± 0.0	7.2 ± 1.2	1.1 ± 0.0	0.4 ± 0.0	0.1 ± 0.0	3.7 ± 0.2	0.7 ± 0.1	0.1 ± 0.0		5.3 ± 1.9	0.8 ± 0.1	5.0 ± 0.3		12.0 ± 0.8	1.1 ± 0.2
2	1251.25	1250	Guvacoline	82	126, 141	4.606	y = −25.21x^2^ + 114.01x + 1.875	0.9997	3.12–100 mg/L	-	-	-	1.1 ± 0.1	-	-	-	0.5 ± 0.2	-	-	-	0.5 ± 0.0	-	-	-	-
3	1279.43		Arecaidine	96	141, 94	4.832	y = 583.58x + 19.175	0.9802	25–100 mg/L	-	-	-	-	>85.49	-	-	>3.23	-	-	-	-	-	-	-	-

mg/g DM for arecoline and mg/100 g DM for guvacoline and arecaidine; RI, retention index experimental; RIdb, retention index database.

## Data Availability

The data presented in this study are available upon reasonable request form the corresponding author.

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
