# Peer review of "Metabolomic Profile of Indonesian Betel Quids"

_biomolecules, 2022, doi:10.3390/biom12101469_

Round 1

Reviewer 1 Report

This is a potential interesting manuscript dealing the analysis of betel quid components in various regions of Indonesia. But some revisions are suggested to improve the manuscript.

1. “Variations are also known to exist including the addition of betel stem and inflorescence (BS and BI), and various additives [1, 2] .” References 1-2 not really revealed the importance in addition of betel stem and inflorescence. Other papers are better to be included (e.g., IARC 2004; Oral Oncol. 2001 Sep;37(6):477-92; Lancet 1992 Sep 5;340(8819):577-8.)

2. What’s the major difference of this paper from the paper (Sci Rep. 2020 Oct 1;10(1):16254) published recently by the same authors?

3. Page 2, while “……..Indonesian BQ mixtures [7], the highest arecoline concentration was observed in the nuts and BQ mixtures from West Papua; BQ mixtures from other regions of the country showed considerably lower concentration of arecoline “. These results seem linkage the association between arecoline and oral premalignant lesions?

4. Whereas arecoline is perhaps to be one contributing component of oral premalignant lesions, the etiologic mechanisms seem not well identified?  Some references may be raised for discussion (Prague Med Rep. 2020;121(4):209-235; Int J Mol Sci. 2020 Oct 30;21(21):8104; J Oral Pathol Med. 2020 Apr;49(4):305-310.).

5. Page 2, line 50-54: The sentence “The mutagenic effects of constituents and extracts of the AN have been studied extensively in vitro and in vivo [8-12]. Four major AN alkaloids, namely arecoline, arecaidine, guvacine, and guvacoline have been isolated and suggested to play a key role in oral carcinogenesis.” although generally acceptable, can also give some bias. Reactive oxygen species, and nitrosamines (the study group of Prof. Nair or Prof. Bartsch) should be also mentioned for avoid of misleading.

6. line 18: guvacine hydrochloride were…..

7. “Discussion” The involvement of safrole, and especially hydroxychavicol in oral submucous fibrosis and oral cancer can be briefly discussed and referenced.

8. In addition to betel inflorescence and betel leaf, whether safrole is really present in areca nut?

Reviewer 2 Report

improved much

Author Response

No reply/comments needed

Reviewer 3 Report

The authors studied the Metabolomic profile of Indonesian betel quids as Indonesia is one of the largest producers of AN worldwide. I congratulate the authors for their sincere effort …but a lot more value may be added to their article by adding the forthcoming points.

Authors have tried to explain the prevalence of oral potentially malignant disorders (OPMDs) among different regions of Indonesia based on this profile. West Papua has been shown to have a much higher prevalence of OPMDs compared to Jakarta and West Java. (In Line 47) Notably, the highest arecoline concentration was observed in the nuts and BQ mixtures from West Papua (and this might be the reason? – add references for this).

Authors opine "addition of slaked lime, namely calcium hydroxide, facilitates hydrolysis of arecoline to arecaidine, which leads to amplified fibroblast proliferation, increased collagen formation, and potentially to an increased risk of oral cancer”. - these lines have been cited many times; authors may retain theses but may I suggest adding these….

1.     Jaunberzins A, Gutmann JL, Witherspoon DE, Harper RP. Effects of calcium hydroxide and transforming [correction of tumor] growth factor-beta on collagen synthesis in subcultures I and V of osteoblasts. J Endod. 2000 Sep;26(9):494-9

2.     Jaunberzins A, Gutmann JL, Witherspoon DE, Harper RP. TGF-beta 1 alone and in combination with calcium hydroxide is synergistic to TGF-beta 1 production by osteoblasts in vitro. Int Endod J. 2000 Sep;33(5):421-6. 

3.     Donoghue M, Basandi PS, Adarsh H, Madhushankari GS, Selvamani M, Nayak P. Habit-associated salivary pH changes in oral submucous fibrosis-A controlled crosssectional study. J Oral Maxillofac Pathol 2015;19:175–81

4.     Kuo TM, Luo SY, Chiang SL, Yeh KT, Hsu HT, Wu CT, Lu CY, Tsai MH, Chang JG, Ko YC. Fibrotic Effects of Arecoline N-Oxide in Oral Potentially Malignant Disorders. J Agric Food Chem. 2015 Jun 24;63(24):5787-94

Articles 1 & 2 slaked lime (Chemically Calcium hydroxide) may chemically collaborate with Tgf-beta to induce fibrosis. Thus, addition of slaked lime will cause more fibrosis by this mechanism also. Article 3 states that an alkaline environment may promote fibrosis by changing the fibroblast phenotype. Article 4 paraphrases and add this ..“Arecoline N-oxide, a metabolite of areca nut alkaloids, which has been identified in animal urine, has been shown to induce mutagenicity in bacteria. In this study, it was found that its protein adduct could be detected in oral keratinocytes treated with areca nut extract. Increased collagen expression and severity of squamous hyperplasia were observed in arecoline N-oxide treated mice. In cultured oral fibroblasts, arecoline N-oxide showed stronger effects on the increase of fibrotic related genes including TGF-beta1, S100A4, MMP-9, IL-6, and fibronectin and a decrease of E-cadherin as compared with arecoline. Finally, arecoline N-oxide stimulation effectively increased the DNA damage marker, gamma-H2A.X, both in vitro and in vivo. Taken together, these results indicate that arecoline N-oxide shows a high potential for the induction of OPMD.” (Add these to discussion).

However, the role of slaked lime as an independent carcinogenic agent should be highlighted in the manuscript

[

1.     Thomas SJ, MacLennan R. Slaked lime and betel nut cancer in Papua New Guinea. Lancet. 1992 Sep 5;340(8819):577-8.

2.     Nair UJ, Friesen M, Richard I, MacLennan R, Thomas S, BartschH. Effect of lime composition on the formation of reactive oxygen species from areca nut extract in vitro. Carcinogenesis. 1990;11: 2145–8.

3.     Lewis A, Hayashi T, Su TP, Betenbaugh MJ. Bcl-2 family in interorganelle modulation of calcium signalling; roles in bioenergetics and cell survival. J Bioenerg Biomembr. 2014;46:1–15.

4.     Hennings H, Kruszewski FH, Yuspa SH, Tucker RW. Intracellular calcium alterations in response to increased external calcium in normal and neoplastic keratinocytes. Carcinogenesis. 1989;10: 777–80.

5.     Mukhopadhyay S, Munshi HG, Kambhampati S, Sassano A, Platanias LC, Stack MS. Calcium-induced matrix metalloproteinase 9 gene expression is differentially regulated by ERK1/2 and p38 MAPK in oral keratinocytes and oral squamous cell carcinoma. J Biol Chem. 2004;6:33139–46.]  (Add this to discussion).

In line 60 authors opine that “ Betel SI containing safrole is likely to be carcinogenic 60 [17-19], however it is not known whether....  however, several articles show the anticarcinogenic activity of safrole [1 -7]. Cite and add these to the discussion..

1.              Zhao BX, Du AY, Miao JY, Zhao KW, Du CQ. Effects of novel safrole oxide derivatives, 1-methoxy-3-(3,4-methylenedioxyphenyl)-2-propanol and 1-ethoxy-3-(3,4 methylenedioxyphenyl)-2-propanol, on apoptosis induced by deprivation of survival factors in vascular endothelial cells. Vascul Pharmacol. 2003 Oct;40(3):183-7.

2.              Zhao J, Miao J, Zhao B, Zhang S, Yin D. Suppressing Akt phosphorylation and activating Fas by safrole oxide inhibited angiogenesis and induced vascular endothelial cell apoptosis in the presence of fibroblast growth factor-2 and serum. Int J Biochem Cell Biol. 2006;38(9):1603-13.

3.              Catalán LE, Villegas AM, Liber LT, García JV, Fritis MC, Altamirano HC. Synthesis of nine safrole derivatives and their antiproliferative activity towards human cancer cells. J Chil Chem Soc 2010;55(2):219–222.

4.              Madrid Villegas A, Espinoza Catalán L, Montenegro Venegas I, Villena García J, Carrasco Altamirano H. New catechol derivatives of safrole and their antiproliferative activity towards breast cancer cells. Molecules. 2011;16(6):4632-41.

5.              Yu FS, Yang JS, Yu CS, Lu CC, Chiang JH, Lin CW, Chung JG. Safrole induces apoptosis in human oral cancer HSC-3 cells. J Dent Res. 2011;90(2):168-74.

6.              Hung TY, Chou CT, Sun TK, Liang WZ, Cheng JS, Fang YC, Li YD, Shieh P, Ho CM, Kuo CC, Lin JR, Kuo DH, Jan CR. The Mechanism of Safrole-Induced [Ca²⁺]i Rises and Non-Ca²⁺-Triggered Cell Death in SCM1 Human Gastric Cancer Cells. Chin J Physiol. 2015;58(5):302-11.

7.                   Eid AM, Hawash M. Biological evaluation of Safrole oil and Safrole oil Nanoemulgel as antioxidant, antidiabetic, antibacterial, antifungal and anticancer. BMC Complement Med Ther. 2021;21(1):159.
